# Drone Observation for the Quantitative Study of Complex Multilevel Societies

**DOI:** 10.3390/ani13121911

**Published:** 2023-06-08

**Authors:** Tamao Maeda, Shinya Yamamoto

**Affiliations:** 1Wildlife Research Center, Kyoto University, Kyoto 606-8203, Japan; 2Research Center for Integrative Evolutionary Science, The Graduate University of Advanced Science (SOKENDAI), Hayama 240-0193, Japan; 3Institute of Advanced Study, Kyoto University, Kyoto 606-8501, Japan; shinyayamamoto1981@gmail.com

**Keywords:** drone, UAV, orthomosaic, multilevel society, complex society

## Abstract

**Simple Summary:**

Multilevel societies are the most complex social structures in animals. They are social structures with nested levels of social organization, where stable “core units” gather to form higher-level groups. Such multiple-unit groups usually become very large (usually 100–1000 individuals), making it difficult to conduct observations. Most previous studies lack numerical definitions and descriptions; thus, it is largely unknown how and why animals, including humans, form multilevel societies. Aerial observation from unmanned aerial vehicles (drones) is a powerful method of capturing the spatial structures and movements of large animal groups. Recent advances in drone technology and its application have led to new discoveries, such as the spatial structure of multiunit groups and the underlying mechanisms of collective behavior. We review how previous studies have utilized drones and how drone applications have contributed to the understanding of multilevel societies. We will further discuss the potential opportunities and challenges with regard to drone observations in future studies.

**Abstract:**

Unmanned aerial vehicles (drones) have recently been used in various behavioral ecology studies. However, their application has been limited to single groups, and most studies have not implemented individual identification. A multilevel society refers to a social structure in which small stable “core units” gather and make a larger, multiple-unit group. Here, we introduce recent applications of drone technology and individual identification to complex social structures involving multiple groups, such as multilevel societies. Drones made it possible to obtain the identification, accurate positioning, or movement of more than a hundred individuals in a multilevel social group. In addition, in multilevel social groups, drones facilitate the observation of heterogeneous spatial positioning patterns and mechanisms of behavioral propagation, which are different from those in a single-level group. Such findings may contribute to the quantitative definition and assessment of multilevel societies and enhance our understanding of mechanisms of multiple group aggregation. The application of drones to various species may resolve various questions related to multilevel societies.

## 1. Introduction

How have the various animal societies evolved? What is the mechanism for maintaining social groups? Various studies have been conducted to answer these questions, including field observations and laboratory experiments using captive animals. In-lab studies are extremely important for testing hypotheses because they can be controlled experimentally; however, not all animals exhibit natural sociality in captivity. Quantitative studies of animals in the wild are also essential for comparing various types of social systems, which leads to an understanding of the proximate and ultimate factors that shape animal societies.

Recent advances in analysis and observation techniques, such as network analysis and global positioning system (GPS) tags/logger, have contributed to advances in the study of animal societies in the wild [1]. The use of unmanned aerial vehicles (UAVs) or drones is applicable to wildlife studies. In the 2000s, technological advances and the deregulation of GPS accessibility led to the commercial release of small general-purpose drones. Since then, the use of drones has rapidly increased in various research fields [2]. The advantage of drones in wildlife studies is that they can observe the behavior and movement of numerous animals simultaneously with high resolution. It is even possible to obtain images of a few centimeters/pixels with errors of <10 cm in a geographical location [3]. Drones do not require animal capture and they are less costly compared to alternative methods, such as GPS tags and loggers. In addition, they can be used to survey locations that are difficult to access on foot [4]. Alternatively, flight time, flight range, and techniques for detecting animals from aerial images have often been limited to observation and data collection [5]; however, these problems are gradually improving (see Section 5.2 for further discussion on drone problems).

To date, drones have mostly been used for studies on kinetic interactions in a single-level group [6,7,8,9,10] or population survey/monitoring [4,11,12]. There are very few examples of their use in complex social communities. Social complexity is generally determined by the number and diversity of social relationships within a society [13,14]. Relationships within a complex society are often determined by various factors, such as kinship, past interactions, and dominance [15,16,17]. Quantifying such complicated relationships requires extensive social data, which limits the feasibility of conducting large cross-species comparisons. In addition, a large number of social relationships implies a large group size, which makes it difficult to conduct observations. Drones are suitable for observing large animal groups [1], and their high-resolution images enable the detection of social events, the identification of individuals and the estimation of individual properties such as sex, size and age [18]; thus, they have the potential to study complex animal social groups.

In this review, we focus on a multilevel society, as a representation of complicated social structures. A multilevel society is a social structure with nested levels of social organization. Individuals are structured into stable unit groups that preferentially associate with other units to form a higher level of social organization [19]. It is considered to be among the most complex social structures in animals [20] because it enables animals to retain social relationships with more individuals than in a single-level society [21] (usually a group consists of more than a hundred individuals, even up to thousands [19,22].) Social relationships vary across social levels; for example, in human hunter–gatherer societies, it has been reported that different social levels are known to show different levels of food sharing rates, and such cooperation has been affected by kinship and non-kin bonds [23]. In addition, multilevel societies have been found primarily in large mammals, probably because the social structure demands high cognitive ability, which also makes it difficult to observe them in captive or experimental conditions (for a more detailed review of multilevel societies, see [19,20]). Previous reviews on multilevel societies mainly focused on their characteristics, evolutionary history and function; therefore, to the best of our knowledge, this is the first review that discusses multilevel societies based on methodological aspects. We first introduce the difficulty of observing a multilevel society using conventional methods, then we discuss several drone applications that could be used to address such challenges, and describe the potential application of drones in further studies.

## 2. Challenges of Studying Multilevel Societies

The function, mechanism of aggregation, and evolutionary history of multilevel societies remain poorly understood because of the difficulty in quantitative studies [24]. As direct social interactions, such as grooming, are quite rare at the inter-unit level [25], researchers usually rely on proximity data to detect each level of the hierarchy or measure social relationships by social network analyses, which require that we define proximity with a certain threshold of inter-individual distance (e.g., 100 m) [26,27,28,29,30,31]. However, most studies failed to provide objective reasons for why such a threshold distance is meaningful for focal animals. It is possible that the lack of continuous distance data may result in failure to properly measure social relationships. If a large threshold is selected, the network will be too dense, and units will be difficult to distinguish; conversely, if the threshold is too small, the network will be too sparse to detect any social relationship. In addition, the lack of reasonable standards to measure social relationships causes difficulties in conducting inter-population and inter-specific comparisons. In fact, the definition of each hierarchy has often been confusing among different studies [25]; however, there is still no uniform definition [19].

In addition, limited information in the data also caused bias in the study themes. One of the least understood areas in multilevel societies is collective behavior across unit groups. Researchers have identified several rules behind dynamic swarming behaviors that are common to various taxonomic groups. However, previous studies have primarily discussed collective behavior, focusing on a single group of fish, birds, and invertebrates in experimental systems. In multilevel societies, units show a high tolerance to different units [32], and they often gather to create a temporal multiple-unit group [27,33,34]. Although a few previous studies on collective movement/decisions in multilevel societies exist, they have used data obtained from ground-level observations, thus failing to consider inter-individual distance or the spatial structure of a group [35,36], which is among the most fundamental factors for collective decision making.

Several studies have implemented GPS tags instead of direct observations and have succeeded in obtaining accurate distances [34,37]. Although GPS tags are a powerful methodology, the problem is the difficulty of tagging numerous individuals. In a multilevel social group, it is necessary to obtain both intra- and inter-unit (and intra- and inter-sexual) associations to understand stratified relationships; thus, researchers need to implement more GPS tags than in a single-level society. For collective behavioral studies, much more positioned individuals are required to accurately model movement. In fact, no study has measured collective behavior at the inter-unit level using GPS (although there are several intra-unit level studies: [37,38]). Although GPS tags have substantial advantages in obtaining positional data in a wider range of geographical areas, they are not suitable for investigating the spatial structure of multiunit groups and their collective movements.

In the next two sections, we focus on how drones contribute to: (1) The detection and measurement of social relationships in a multilevel society; and (2) Understanding the collective behavior of a multilevel social group.

## 3. Using Drones to Understand Group Structure and Measuring Social Relationships in a Multilevel Society

### 3.1. The Necessity of Quantitative Assessment of the Multilevel Social Group Structure

Although the definition of a multilevel society more or less varies among studies, there are three main conditions by which a multilevel society is recognized; (1) The existence of a stable core unit: members of the same unit are physically closer to each other than to individuals who are not [21,26,28,30,32]. Membership is usually very stable, and the composition does not change for months or even several years [27,39,40]; (2) Aggregation of multiple units: different units usually exhibit fission–fusion characteristics and gather to create a temporal multiple-unit group [27,33,34]; (3) Social relationships among units: associations among units are determined not only by environmental, but also by social factors. Certain units are more likely to associate with or avoid each other than for randomly interacting situations to occur [26,32,40]. When animals have more than two levels of social hierarchy, Conditions 2 and 3 are also applicable to higher-level social groups, and members of lower-level groups have stronger bonds than those of higher-level groups [25].

As mentioned in the above sections, most of the studies used the association index (AI), the ratio of inter-individual distance that is smaller than a certain threshold, and subsequent cluster analysis to verify that the subject species has a multilevel society [26,29,40,41]; however, the threshold for ascertaining AI is often determined without any objective basis (see studies on golden snub-nosed monkeys [34]). In several cases, the results of clustering are supported by behavioral data, such as movement or membership stability [26]. However, the lack of a unified definition makes inter-population and species comparisons difficult. Drone observations us can help obtain accurate positional data and enable a more quantitative assessment of the population structure of multilevel societies, which may facilitate more rigorous detection and comparisons among studies.

### 3.2. How to Apply Drone Methods

To date, there is only one case in which a drone was used to examine whether there is a multilevel society and to measure its social relationships [42]. Regarding the horses included in this study, more than 100 individuals were scattered over an area of several hundred meters. To capture all of them in one frame, a drone needs to fly at an altitude of over one hundred meters, which is too high for individual identification, and the distortion of the lens makes it difficult to calculate the exact distance. To solve these problems and to accurately calculate the identified positions, the present study applied the orthomosaic technique (Figure 1). An orthomosaic is a non-distorted map-like image created by connecting a series of successive photographs. Detailed 3D shape recovery is performed by: (1) Automatic acquisition of feature points; (2) Camera position and orientation estimation and calculation of three-dimensional coordinates of feature points; (3) Point cloud generation using multiview image measurement; (4) Automatic surface shape modeling [43,44,45]. Each image was captured at a low altitude of approximately 40–50 m, so that the resolution was sufficiently high for individual identification. The horse population was primarily identified by direct observation from the ground, and their features were matched with the aerial images. In addition, each pixel in the orthomosaic is embedded in the GPS information, which enables accurate distance measurements. This technique has been primarily used for vegetation [46,47,48], topographic surveys [49,50] and population surveys of wild animals [4,11,51]. Thus, it was the first attempt to apply it to animal social studies. Combined with individual identification, this orthomosaic method has the advantage of being able to monitor social associations in the long term by adding individual information on positional data, whereas previous drone studies could not track individuals.

Using this method, this drone study also revealed the spatial structure characteristics of a multilevel society. The inter-individual distance in a single-level animal group, such as bird flocks, fish swarms and angulate herds, is generally described using an attraction–repulsion model [53,54,55]. Individuals try to maintain almost identical nearest-individual distances, where attraction and repulsion forces are in equilibrium. Indeed, inter-individual distances are not always homogeneous due to temporal fluctuations and various other factors (social relationships [10,56], positions in a group [57], individual properties [58], and environmental factors [59,60]), but such effects are often continuous, thus the overall distance distribution is expected to show unimodality [30]. However, in a multilevel society, the same unit members showed higher cohesion and different units were scattered at relatively farther distances (yet closer than randomly positioned points), and such a structure was represented by the bimodal structure in the inter-individual distance histogram. This structure in a multiunit group has been suggested in a study on red uakari [30]; however, this is the first time that it has been verified using quantitative data.

This study indicates that different thresholds are necessary for the evaluation of intra- and inter-unit relationships. In measuring intra-unit relationships, it would be appropriate to set the threshold around the first peak of the histogram of inter-individual distances, whereas it should be around the second peak for the inter-unit relationship to be measured; otherwise, the network can become too sparse. Horses form a two-layered multilevel society (only core unit and second-level social organization), but for a three- or four-layered society, such as the gelada society, it is even more difficult to evaluate every type of proximity relationship with a single threshold. Obtaining continuous interindividual distances could provide objective reasoning for selecting the threshold and enable a more accurate assessment of stratified relationships in a complex society. Accurate inter-individual distance data are difficult to obtain by direct observations from the ground; thus, aerial observations from drones will contribute to a more precise evaluation and detection of multilevel societies.

Furthermore, the aerial observation provided positional information on individuals and units in a multiple-unit group. It would have been feasible to identify to whom an individual was close using the previous methodology; however, it was impossible to know where individuals were located in a group. The individual was positioned heterogeneously in a multi-unit group (Figure 2), and this also raises a question about their collective behavior because most of the models for such studies assumed a homogeneous inter-individual distance in a group. How do multilevel social animals maintain the cohesion of a specific group structure?

## 4. Collective Behavior in a Multilevel Society

To understand the mechanism of cohesion, it is essential to investigate the group structure and the group coordination behavior. Groups of animals often exhibit coordinated behavior among individuals, and previous studies have revealed that the central tenet of collective behavior is that simple repeated interactions between individuals can produce complex adaptive patterns at the group level [53,61]. In several species, from simple multicellular organisms (Placozoa) to vertebrates, such as fish, birds, and even human pedestrians, it is believed that local interactions among several nearest-individuals are sufficient to explain the emergence of global patterns [61].

Drones have also been used in the collective behavioral studies of terrestrial animals. A famous example is the study of the collective migration of caribou [10]. Drones succeeded in obtaining the movement of wild caribou using videos with automatic tracking based on machine learning, and a model was created to explain their movement patterns and local interactions among individuals. Auto-tracking of terrestrial animals has been conducted in various terrestrial animals such as plains zebras [62], horses [63], and wildebeest [64] (for a more general review on drone utilization for the investigation of group coordination, see [1,5,65]).

However, most previous studies have focused on collective behavior in a single group of animals, and there is a paucity of knowledge about group coordination in a multilevel society. One of the major differences in collective behavior in a multilevel society is that they interact with multiple units. In a single-level society, the ingroup/outgroup distinction is clear: all the group members are ingroup and the others are outgroup; therefore, researchers need to consider only a single type of interaction among group members. However, in a multilevel society, this discrimination is more complex. A member of a different unit group should be an outgroup from the viewpoint of units, while s/he could be an ingroup at the level of a higher-level group. How do animals recognize different units and adjust their behavior? Can we apply the conventional model to multilevel societies? In this section, we discuss: (1) The characteristics of collective behavior in a multilevel society; and (2) How aerial observation using drones can help develop studies on collective behavior in a multilevel society.

### 4.1. Previous Studies of Collective Behavior in a Multilevel Society

To date, most of the studies on group movement in a multilevel society have focused on the traveling order. Most of them described and analyzed only the order of departure or arrival because the data were collected by direct observation or video recording from the ground, and it was difficult to obtain positional data [35,36,66]. It is not yet clear whether there is a distinctive rule in collective behavior when compared with single-level societies. Some argue that multilevel societies have no particular means for collective decision mechanisms. The properties of departure (sex bias of initiator and individuals in the front, decisive factor of departure, and correlation between dominance rank and leadership) in Guinea baboons living in a matrilineal multilevel society were more similar to single-level and matrilineal baboon species, such as chacma, yellow, and olive baboons, rather than multilevel and patrilineal hamadryas baboons [36]. The results suggest that their movement pattern is more influenced by matrilineality/patrilineality, not by whether they have a multilevel society.

Alternatively, experimental studies on collective decisions in human societies suggest that multilevel social groups can have a distinctive advantage over a single-level society when finding better solutions to a complex problem [67]. Fully connected groups tend to be stuck with local optima; on the other hand, in a multilevel society, each unit can move freely to search for their own answers, and units are loosely connected to share different information, allowing them to combine local solutions and reach more optimal and complex answers. Derex and Boyd further argue that such a decision-making system could accelerate the accumulated cultural evolution, which is why humans have diverse and complex cultures compared to other apes who do not have a multilevel society [67,68]. In this study, information propagation was completely controlled by experimenters; thus, the verification of the hypothesis must be conducted in natural social settings.

### 4.2. How to Apply the Drone Method

Recently, some researchers have started using drones to reveal the characteristics of collective behavior in a multilevel society of equine species, domestic horses (*Equus caballus*) [52], and Przewalski’s horses (*Equus ferus przewalskii*) [69].

The study of domestic horses investigated behavioral synchronization within and across units as a mechanism of group coordination using orthomosaics, which were taken at 30 min intervals. Combined with accurate positional data, the behavioral states (resting or active) of horses were coded. This study further implemented an agent-based model to explain synchronization in resting behavior, and the simulations suggested that resting and non-resting behavioral states are synchronized both within and across units in a multilevel society [52]. The same unit members spent most of their time within 15 m (10 body lengths) of each other; however, the closest distance between different units was approximately 40 m [42]. This suggests that there is an interaction between individuals at a much greater distance and in greater numbers than previously thought.

A study on Przewalski’s horses used drone videos and recorded the movement of approximately 240 individuals (Figure 3). Their migration behavior showed a strict hierarchy of leaders and followers within and between units [69]. They assumed that only the leader of each unit is responsible for inter-unit behavioral propagation, whereas others only recognized the behavior of the same unit members. The simulation suggested that this modular leadership network could propagate behavior more effectively than a single-level network, and the proportion of leaders in horse multiunit groups almost matched the simulated the network with optimal behavioral propagation efficiency [69]. This study suggests a functional advantage of heterogeneous behavioral propagation in a multilevel society.

The use of videos enabled a higher resolution in the time frame compared to orthomosaics, following the movement of the horse multiunit group. However, the studies did not track individuals; thus, kinetic interactions among individuals were not analyzed. In addition, they did not provide individual identification, although they succeeded in identifying each unit and unit leader and were unable to show the social context of leadership emergence in a horse herd. The absence of individual identification would be due to the higher altitude of drone flights, and less physical variations in Przewalski’s horses compared to domestic horses. The combination of orthomosaic techniques and video tracking may enable researchers to obtain both high-resolution images from low altitude for individual identifications and movement data from high altitude (further discussion on this problem is on Section 5.2).

Drones made it possible to simultaneously observe all individuals in the population and measure proximity relationships, which contributed to the discovery of new facts. Both studies showed inter-unit propagation of movement/behavior, and the strength of synchronization was greater within units, implying that horses distinguish unit members from others. The study by Maeda et al. suggests that collective behavior in multilevel societies requires a different model from that in single-level societies. It was believed that the establishment of multilevel societies may require relatively high cognitive abilities [70] (although they have also been found in birds with a small brain [26]), and recent studies have revealed highly sophisticated social cognition in domestic horses [71,72,73,74], which may enable them to form a multilevel society. Further investigation of the details of cognitive requirements for the collective behavior of a multilevel social group will provide more insight into the cognitive constraints on the evolution of a multilevel society.

## 5. Possibilities for Future Studies

### 5.1. What Can Drones Contribute to Future Studies?

There are still only very few studies of multilevel societies using drones; however, they have a potential to lead to new discoveries related to various scientific questions. A previous study succeeded in verifying the presence of a multilevel society combining aerial images and social network analysis. The utilization of an agent-based model has led to new insights into collective behavior among multiple-unit groups. Moreover, there is also the possibility of combining positional information with other data, such as relatedness, to further look into the functions of a multilevel society. We would propose a framework for drone utilization in a multilevel society (Figure 4). Given that inter-individual distances and movements are intrinsic information in any animal group, drone observation enables a more quantitative definition of multilevel societies and an evaluation of movement characteristics in various species, which facilitate comparative studies, which would allow us to discover similarities and differences in spatial structure and behavioral propagation patterns among species and populations. “Multilevel societies” are composed of a variety of social structures and evolutionary process. For example, the structure of units varies from harems to matrilineal and multimale–multifemale groups [19]. African papionines developed multilevel societies through a process in which large multimale–multifemale groups were divided internally into one-male units, whereas Asian colobines have developed them through semi-permanent aggregation of multiple units [34,75]. Moreover, there should also be inter-population variations owing to environmental differences, although almost no studies have compared them quantitatively. For example, the social structure of killer whales is known to be influenced by prey type; the fish-eating population shows a multilevel society and philopatry, whereas the group size of the mammal-eating population is generally small [76]. We expect that such variations can also occur gradually according to the environmental gradient. The exploration of the effects of such interspecific and inter-population variations on spatial structure and group behavior would provide a deeper understanding of the mechanisms that maintain complex groups and their functions. For these comparisons, high-resolution quantitative data collected using drones would be very useful.

Little is known about even the basics of collective behavior in multilevel societies. Although it has been shown that behavior propagates even between different units, the extent to which individuals recognize other units and whether there is a difference in the process of behavior propagation within and between units remains unresolved. In a study of feral horses, there was no clear leader–follower relationship in behavior propagation [52], whereas another study on Przewalski’s horses assumed a strict hierarchy where only a representative, or a leader, of a unit was responsible for the inter-unit interactions [69]. They both assumed that behavior is transmitted on an individual–individual basis and that inter-unit transmission occurs continuously and linearly; however, it is also possible that inter-unit behavioral transmission occurs non-sequentially. For example, it is possible that individuals do not always recognize the behavior of other groups; however, they can react only when they stray too far from other units or when the behavior within another unit is fully synchronized. Further application of drones is necessary to investigate the features of collective behavior in multilevel societies.

As Derex and Boyd [67] demonstrated, it is also important to examine whether collective behavior and collective decisions in multilevel society groups are more adaptive to complex problems than in single-level societies. It has been believed that multilevel societies can increase foraging efficiency and have less predation pressure/conspecific harassment simply because of their large group size [75]; however, the influence of the structural features of the social network has not been considered. Ozogány and Viscek [69] observed that a modular structure in a horse herd has the benefit of more effective leadership compared to that in a non-modular structure, as information is transferred relatively quickly among a few leaders, and leaders transfer information to followers in sub-groups. Thus, the further application of drones may contribute to the understanding of structural advantages in a multilevel social group.

To date, studies of drones in multilevel societies have been limited to those using only distance or movement data; however, they have the potential to be combined with other information. To further understand the social system, it is essential to compare spatial data to social and kinship data by combining direct observation and genetic analysis, as has been implemented in some drone studies in a single group [9]. Furthermore, it is possible to observe social behavior, not just movement. Recently, several software packages that can estimate the posture of animals in aerial images and videos using deep learning have been released [62]. It enables a more detailed understanding of the trajectory of the movement and even the eyesight of animals, thus facilitating a more precise understanding of a certain behavior by observing who approached whom, from what distance, and in what context.

### 5.2. Challenges in Drone Observation and Their Possible Solutions

Although drones are an impressive technology, several problems still need to be addressed, and such problems can be solved by combining them with other technologies and analytical methods. One major issue with drones is that they are difficult to observe in non-open areas and under conditions of poor visibility (nighttime, fog, etc.). An increasing number of ecological studies have implemented drones deployed with a thermal infrared camera to observe animals in forests or during the night [12,77,78]. As the resolution of the image is usually very low (usually around 600 pixel) compared to RGB cameras [65] and it may be difficult to identify individual animals, further efforts are needed to utilize thermal cameras for behavioral observation. Another problem is the short observation time and narrow observation range. Regarding the observation time, an increasing number of models can fly for longer periods of time, and efforts are also being made to extend the observation time by switching between two drones. From 100 m altitude, for example, the area that can be observed is at most 200 m wide when drones have viewing angles of 90° (for example, DJI Phantom and Mavic series, models often used in observation, have a viewing angle of 80–90° [79]). It is likely that higher-level groups are spread in larger areas [26,34,42]; thus, it is necessary to develop a method to combine with other methodologies, such as GPS tagging, or use multiple drones simultaneously to collect data from a wider range.

In addition to hardware problems, individual identification can also be an issue for aerial observations. Studies of sociality usually require individual identification and long-term observations. Even for collective behavioral studies, it is preferable to obtain individual properties because social relationships and social status are known to have measurable impacts on individuals’ decisions to move [80]. Most drone studies did not implement identification as they focused on collective movements; however, recently, several studies have succeeded in individual identification from aerial images using machine learning based on pattern, color, or scratches on the animal bodies [81,82,83,84,85]. In studies of multilevel societies, individual identification has been achieved in horses with the support of ground-level observations, but not in Przewalski’s horses. The altitude from which the animal is observed is a key factor influencing the feasibility of individual identification. When viewed from a high altitude, the number of pixels for each individual inevitably becomes small, making it difficult to distinguish them from each other (although it depends on the image resolution of a drone camera). Nevertheless, it is difficult to observe large herds from low altitudes. In fact, photographs from low altitudes (around 5–40 m) are often used for individual identification [42,81,82,83], while drones are often operated at 70 m or higher altitude in collective behavioral studies [8,10,63]. As mentioned in Section 3, the orthomosaic technique provides a solution for this dilemma, as it makes it possible to conduct observations at lower altitudes covering a wide range. When recording movements, it may be worthwhile to combine video imaging from a high altitude with individual identification through orthomosaics from a low altitude.

Although orthomosaic techniques may facilitate identification from aerial images, it may still be difficult to use them with animals that have few physical features. This is definitely a general problem for animal social studies in general. Nevertheless, animals without patterns are often identified using their faces [86], although faces cannot be captured well from a drone, so identification from aerial images could be further challenging. The number of studies on image-based identification using deep learning is increasing, and it is being implemented in wildlife studies [86,87]. Sometimes deep learning even outperforms human identification [87]; therefore, it is highly applicable to aerial images. Image analysis tools for identifying individuals using body information will only need to be tested in future research.

Finally, researchers should always consider the potential stress caused to animals by drone observations. The use of drones is considered to be a less invasive method compared to direct observations [11,65,88]. Nevertheless, an increasing number of studies have also confirmed that the sound and silhouette of drones induce disturbance in animal behavior [5,84,89]. Furthermore, even if animals do not seem to change their behavior, it is possible that they experience psychological stress. It has been reported that black bears show an increase in heart rate when drones fly above them, even though no discernible behavioral response was detected [90]. Therefore, it is recommended that drones are flown from the highest possible altitude, which makes individual identification challenging. For example, in a subsequent study of the same black bears, it was found that they became less responsive to drones, and finally, showed no increase in heart rate after 4 weeks of habituation [91]. In drone research, it would be necessary to begin at the highest altitude possible, introduce a habituation period, and then decrease the altitude gradually. In addition, the extent of stress induced by drone observation varies among focal species, e.g., the presence of aerial predators [92,93], and environmental factors, e.g., ambient sound [51,94,95] (for systematic review, see also [96]). The operation of the drones should be carefully planned depending on each species and study site.

## 6. Conclusions

Multilevel societies are yet to be studied sufficiently because of the difficulty of observation. Observations of a multilevel society using drones are only conducted by two research teams; however, they have reported some new facts that could not be found by ground-based observations. Data obtained from drones can be utilized for: (1) Defining a multilevel society; (2) Understanding the collective behavior in a multi-unit group; and (3) Exploring the function or evolutionary reason of a multilevel society by combining it with other social factors. Drone observations should be applicable to many animals, which enables a variety of comparisons among societies, populations, and species, although there are still many challenges hampering its effective utilization. Quantitative comparisons of multilevel societies may lead to further understanding of their functions and evolutionary process. We hope that drones, which can obtain more spatial and movement information than ground-based observations, can be applied in various multilevel societies and contribute to new discoveries.

## Figures and Tables

**Figure 1 animals-13-01911-f001:**
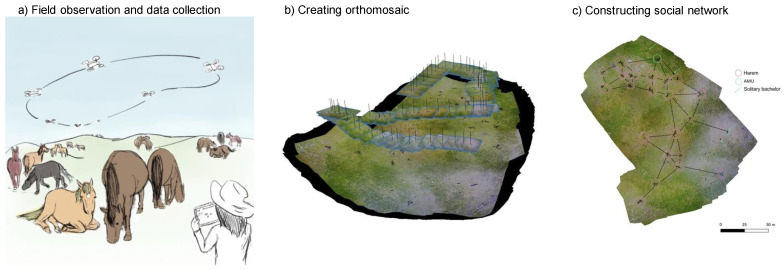
Orthomosaic utilization in a multilevel society. (**a**) A drone flew over the research site and collected successive photos of focal animals. (**b**) The photos were processed to orthomomsaics. (**c**) Sociall networks were constructed based on the positional information obtained from orthomosaics. Reprinted from Maeda et al. [52].

**Figure 2 animals-13-01911-f002:**
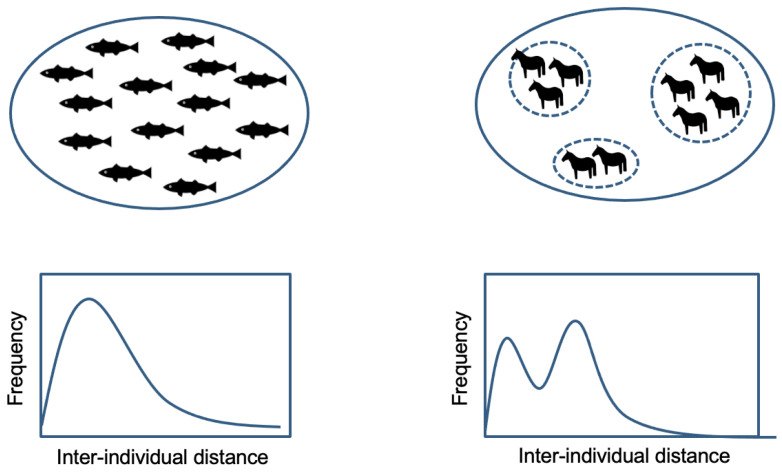
Homogeneous group structure in a single-level group (**left** side) and heterogeneous group structure in a multilevel society (**right** side). The histogram is a predicted structure corresponding to the individual distribution in a group (cf. [42]).

**Figure 3 animals-13-01911-f003:**
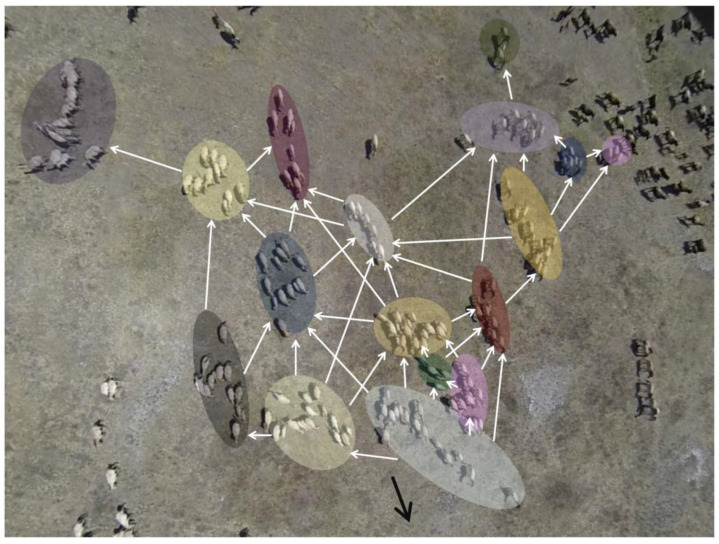
Video tracking of Przewalski’s horse group. Each circle represents units (harems and bachelor groups). The white indicates the order of behavioral propagation through the hierarchical network and the black arrow shows the direction of movement. Reprinted from Ozogány and Viscek [69]. Reproduced with permission from Dr. Katalin Ozogány.

**Figure 4 animals-13-01911-f004:**
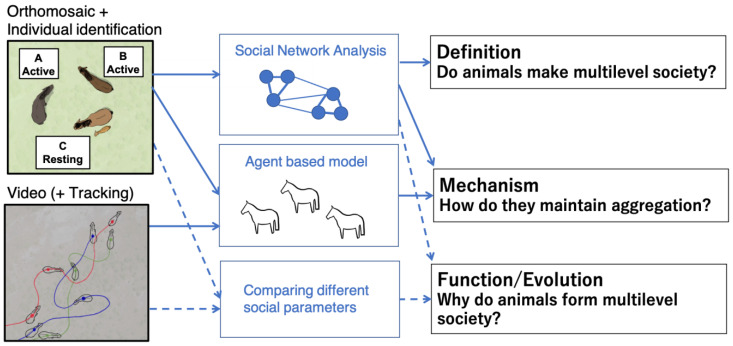
Framework of drone study for a multilevel society. Solid arrows indicate that the methods were already applied in multilevel social studies. Broken lines indicate the methodology that is applied in single-level group and not in a multilevel society.

## Data Availability

Not applicable.

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
