# Peer review of "Drone Observation for the Quantitative Study of Complex Multilevel Societies"

_animals, 2023, doi:10.3390/ani13121911_

Round 1

Reviewer 1 Report (Previous Reviewer 1)

I am appreciative of the time and work that the authors have put into the revision of the manuscript. I am now happy to state that the manuscript has improved substantially and is a pleasure to read. I can therefore wholeheartedly recommend it for publication in Animals.

Author Response

Thank you very much!

Reviewer 2 Report (New Reviewer)

Overall, this is a paper that focuses on the very interesting problem of multilevel societies in animals and the study of these structures using drones. My primary comment concerns the organization of the work itself. The authors have assumed that the work is a review, but the emphasis is misplaced. There is a lack of many papers on this topic, although sometimes the authors cite many papers that are not relevant to the topic of the article. More details follow in the review.

L 44 This paragraph is too short to constitute a paragraph on its own.

Lines 60-70. It is important to at least refer to the use of drones in the study of southern elephant seals. As presented in the paper: Breeding Colony Dynamics of Southern Elephant Seals at Patelnia Point, King George Island, Antarctica in Remote Sensing: “the body surface area of southern elephant seals can be used to assess the development of a given zone, as it is related to the body mass lost by females of a given zone, as it is related to the body mass lost through parturition and latency in females".

Section 3.1.

Lines 135-145. There are stylistic and organizational errors in this paragraph. The authors end the sentence once with a full stop and once with a semicolon. Use semicolons in a serial list. If you follow this rule, then all points (1)-(3) should begin after a semicolon.

Line 160. The first sentence is exclusive of the title and purpose of the work. The same with the sentence in Line 324. If this article is a review concerning multilevel societies in that case, how can a review be done on the basis of few papers? This calls for a rethink and a different positioning of the purpose of the work and its organization.

Lines 161-167. Generally, this is solved by using orthomosaic. I do not understand the discussion presented here.

Lines 168-171. Again, please use semicolons in a serial list.

Section 5.2. One of the main advantages of using drones is the ability to obtain observations over a much larger area, which is not possible to achieve even by a group of observers or in areas that are difficult to access by humans. It is obvious what the authors write about, the possible negative impact, of drones’ noise, but since this is a review, there should be no lack of description of the results of the work The use of drone-based aerial photogrammetry in population monitoring of Southern Giant Petrels in ASMA 1, King George Island, maritime Antarctica from Global Ecology and Conservation Journal, in which the authors showed that SGPs did not show any obvious behavioral signs of disturbance during drone raids, which is inevitable during ground-based observer-led censuses. Moreover, in the context of the results presented in this review, this work provides many relevant elements for discussion in this paper.

Lines 409-412. What is the purpose of citing papers [72-76]? They are irrelevant to the goal of the work.

Author Response

Thank you very much for your comments and recommending the relevant papers. We improved the manuscripts based on your suggestions. We attached the response to each comment.

Round 2

Reviewer 2 Report (New Reviewer)

The paper has to be significantly improved since the first time was review. The authors answered to all of my comments and suggestions. The work done by authors is highly appreciated. In my opinion, the current version of the paper can be published. 

This manuscript is a resubmission of an earlier submission. The following is a list of the peer review reports and author responses from that submission.

Round 1

Reviewer 1 Report

In this manuscript, the authors aim to (1) review the information available about the current use of drones to document and understand the demography, spatial structure, movements, and other collective behaviors in large groups of free-ranging animals, and (2) propose a general framework for the future utilization of drones to inform socio-demographic and behavioral research, particular on multilevel societies in various animal taxa, including humans.

Overall, this is a well-written, well-structured, and relatively informative methodological review that has the potential to contribute to an under-represented line of research.

However, I have one major concern about feasibility that should be addressed before I can recommend publication. A series of requests for clarification should also be addressed. Finally, I flagged a number of grammatical approximations that should be fixed. I hope my comments will help the authors improve the manuscript.

MAJOR CONCERN ABOUT FEASIBILITY REGARDING INDIVIDUAL IDENTIFICATION

Individual identification is a key piece of information for most research combining socio-demographic and behavioral variables, which I believe is the case for studies of multilevel societies. However, the authors did not fully convince me that drone technology as an observational method for complex social structures of animals has (or is about to have) the power to extract these data. Indeed, they kept providing ambiguous and sometimes contradictory statements about this important methodological component of the research.

Here are two examples that illustrate the reasons of my confusion:

L25-26: “Drones made it possible to obtain the identification […] of more than a hundred individuals in a multilevel social group” and LL66-68: “Drones are suitable for observing large animal groups [1], and their high-resolution images enable the detection of social events and identification of individuals” – How do you reconcile these assertive statements with the much less assertive ones on LL160-161 “a drone needs to fly at an altitude of over hundred meters, which is too high for individual identification”, L298: “they did not provide individual identification”, and LL394-395: “individual identification can also be an issue for aerial observations”?

LL300-302: “The combination of orthomosaic techniques and video tracking may enable researchers to obtain identified individual tracking to reveal more detailed interactions and maintain group cohesion in a multilevel society.” (see also LL 402-403: “which may facilitate individual identification” – What do you mean by the modal verb “may”? Is this wishful thinking? What is the degree of confidence that the technology will allow researchers to achieve this result? And as importantly, could you please provide some indications of how this has been occasionally (or will soon be) achieved? Indeed, unless I am missing something important in the power of this technology, it seems a priori difficult to believe that any individual identification can be reliably achieved in any images similar to the one exemplified in Figure 3.

In sum, could you please (1) increase the consistency of your statements regarding the current state of individual identification via drone observation, (2) clarify the tradeoff between the need to fly the drone at an altitude that is high enough to capture all the members of a multilevel society in one frame (see LL160-161) and the need to fly the drone at an altitude that is not too high to provide the image resolution that is required for individual identification (see LL168-170), and (3) provide some information about how drone technology should adjust to the study of different animal taxa within the scope of individual identification?

REQUESTS FOR CLARIFICATION

LL10-11: “hundreds” – What? Individuals? Please specify.

L57: “see Chapter 5 for further discussion on drone problems” – Do you mean section #5 of this manuscript? Please clarify whether you’re talking about the present manuscript or another source, and the latter, which one? (same comment L238: I am not sure about the use of the term “Chapter” for a journal manuscript).

LL320-321: “drone observation should be able to conduct comparative studies” – What do you mean by the modal verb “should”? Is this wishful thinking? When it comes to the comparative method, some of the main methodological issues are inter-researcher reliability and replicability. Please clarify how higher consistency in data collection can be achieved among researchers so that the data obtained by different research teams become comparable.

LL323-324: "Multilevel societies" are composed of a variety of social structures and evolutionary backgrounds.” – Please clarify what you mean by “evolutionary backgrounds” here. Do you mean a history of social interactions among members of these societies? Or do you actually mean the evolutionary processes that led to the establishment of these societies? (similar question for L424: “evolutionary reason of a multilevel society”)

L357: “Ozogány and Viscek” – Citation needed.

L358: “a modular structure in a horse herd has the benefit of more effective leadership” – Compared to what other type of structure? And why?

L388: “viewing angles of 90” – What is the unit of measurement here?

LL 401-403: “the orthomosaic technique makes it possible to conduct observations at lower altitudes, which may facilitate individual identification.” – How do you reconcile this recommendation (for technological reasons) with the one on LL 412-413 (for ethical/animal welfare reasons): “Therefore, it is recommended that drones are flown from the highest possible altitude”?

LL424-425: “Drone observations should be immediately applicable to many animals” – How do you reconcile this statement with the many technological and ethical challenges mentioned in section 5.2.?

MINOR GRAMMATICAL ISSUES

L39: “however, alternatively,” – I would remove “alternatively”.

L40: “studies on animals” – should read “studies of animals”.

L83: “our review is the first time to discuss […]” – Grammatical approximation. Please rephrase.

L86: “propose the potential of drones for further studies” – I think “propose” is not the right verb in this situation. Please rephrase.

L193: “society like gelada” – Grammatical approximation. Please rephrase.

L221: “They” – Not sure who “They” are. Please specify. (same comment L261)

LL251-252: I think that “rather than multilevel” should read “than to multilevel”.

LL268-269: Why “horses” (L268) and then “horse” (L269). Shouldn’t you be consistent (i.e., either both singular or both plural)?

L270: “The study on domestic horses” – Should read “The study of domestic horses”.

LL320-321: “drone observation should be able to conduct comparative studies” – This phrase also contains a grammatical approximation that should be fixed: observation cannot conduct studies; only researchers can.

L354: “than single-level societies” – Do you mean “than in single-level societies”? The latter seems more grammatically appropriate.

Reviewer 2 Report

Drones are a means of image acquisition. The issues highlighted for better knowledge in Complex Multilevel Society come more from image analytic tools default (as clearly stated line 361-370)than from image acquisition. The real awaited innovation is the analyses of video from aerial position.

Nor the tools to obtain such videos nor the data analyses updating were presented or discussed (as expected following the objective statement line 83-84).
 So the title is not an accurate representation of the paper content
The review doesn’t present new data nor synthesis of literature.